# InPars-Light: Cost-Effective Unsupervised Training of Efficient Rankers

**Leonid Boytsov**[*]                                                                    *leo@boytsov.info*
*Amazon AWS AI Labs*
*Pittsburgh*
*USA*
**Preksha Patel**
**Vivek Sourabh**
**Riddhi Nisar**
**Sayani Kundu**
**Ramya Ramanathan**
**Eric Nyberg**
*Carnegie Mellon University*
*Pittsburgh*
*USA*

**Reviewed on OpenReview:** *https://openreview.net/forum?id=sHSKFYyINO*

## Abstract

We carried out a reproducibility study of InPars, which is a method for unsupervised training of neural rankers (Bonifacio et al., 2022). As a by-product, we developed InPars-light, which is a simple-yet-effective modification of InPars. Unlike InPars, InPars-light uses 7x-100x smaller ranking models and only a freely available language model BLOOM, which—as we found out—produced more accurate rankers compared to a proprietary GPT-3 model. On all five English retrieval collections (used in the original InPars study) we obtained substantial (7%-30%) *and* statistically significant improvements over BM25 (in nDCG and MRR) using only a 30M parameter six-layer MiniLM-30M ranker and a single three-shot prompt. In contrast, in the InPars study only a 100x larger monoT5-3B model consistently outperformed BM25, whereas their smaller monoT5-220M model (which is still 7x larger than our MiniLM ranker) outperformed BM25 only on MS MARCO and TREC DL 2020. In the same three-shot prompting scenario, our 435M parameter DeBERTA v3 ranker was at par with the 7x larger monoT5-3B (average gain over BM25 of 1.3 vs 1.32): In fact, on three out of five datasets, DeBERTA slightly outperformed monoT5-3B. Finally, these good results were achieved by re-ranking only 100 candidate documents compared to 1000 used by Bonifacio et al. (2022). We believe that InPars-light is the first truly cost-effective prompt-based unsupervised recipe to train and deploy neural ranking models that outperform BM25. Our code and data is publicly available. `https://github.com/searchivarius/inpars_light/`

## 1 Introduction

Training effective neural IR models often requires abundant *in-domain* training data, which can be quite costly to obtain: For a human annotator, judging a single document-query pair takes at least one minute on average (Han et al., 2020; Kwiatkowski et al., 2019) and a single query may need as many as 50 of such judgements (Buckley et al., 2007).[1] Models trained on out-of-domain data and/or fine-tuned using a small number of in-domain queries often perform worse or marginally better than simple non-neural BM25 rankers

---

[*]Work done outside of the scope of employment.
[1]Robust04 and TREC-COVID collections used in our study have about 1K judgements per query.

Table 1: Average Gains over BM25 for different Models and Training Recipes

| Model name and training recipe | Avg. gain over BM25 | # of "wins" over BM25s ($\leq 7$) |
|---|---|---|
| **Unsupervised**: InPars-based Training Data (three-shot prompting) | | |
| MiniLM-L6-30M (InPars-light) | 1.13 | 7 |
| DeBERTA-v3-435M (InPars-light) | 1.30 | 7 |
| monoT5-220M (InPars) (Bonifacio et al., 2022) | 1.07 | 3 |
| monoT5-3B (InPars) (Bonifacio et al., 2022) | 1.32 | 7 |
| **Supervised transfer learning with optional unsupervised fine-tuning**: transfer from MS MARCO with optional fine-tuning on consistency-checked InPars data | | |
| MiniLM-L6-30M (MS MARCO) | 1.21 | 5 |
| MiniLM-L6-30M (MS MARCO ▶ consist. checked queries) | 1.24 | 7 |
| DeBERTA-v3-435M (MS MARCO) | 1.42 | 7 |
| DeBERTA-v3-435M (MS MARCO ▶ consist. checked queries) | 1.36 | 7 |
| monoT5-220M (MS MARCO) (Bonifacio et al., 2022) | 1.46 | 7 |
| monoT5-3B (MS MARCO) (Bonifacio et al., 2022) | 1.59 | 7 |
| monoT5-3B (MS MARCO+InPars) (Bonifacio et al., 2022) | 1.59 | 7 |

*Consist. checked queries* denotes a set of generated queries filtered out (via consistency checking) using the **DeBERTA-v3-435M** model trained on InPars-generated data.

(Thakur et al., 2021; Mokrii et al., 2021). Good transferability requires (1) large impractical models (Rosa et al., 2022; Ni et al., 2021), and (2) datasets with large and diverse manually annotated query sets.

A recent trend to deal with these problems consists in generating *synthetic in-domain* training data via prompting of Large Language Models (LLMs). This trend was spearheaded by a recent InPars study (Bonifacio et al., 2022). However, proposed solutions are not cost effective because they require either querying the costly generative models or training impractically large rankers. Although follow up studies, in particular by Dai et al. (2022), claimed improvements upon InPars, these improvements were not demonstrated under the same experimental setting. Moreover, researchers used primarily proprietary LLMs whose training procedure was not controlled by the scientific community. Thus, outcomes could have been affected by data leakage, i.e., training of models on publicly available and popular IR collections whose copies could have ended up in the LLMs training data. As such, there is an important question of whether we can obtain comparable or better results using *only* open-source models trained by the scientific community.

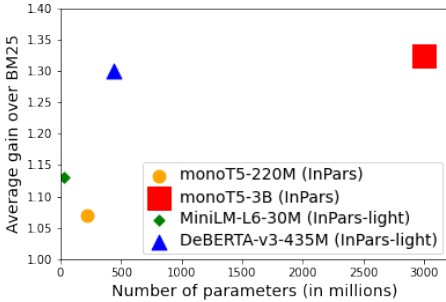

Figure 1: Average relative improvement over BM25 for different model types/sizes and training recipes. Higher and to the left is better. We compare InPars with InPars-Light for the unsupervised training scenario, where training data is generated by an LLM using a three-shot prompt.

This study is driven by two high-level inquiries: (1) Does InPars work? (2) Can it be made more accurate and cost effective? To address these inquiries, we carry out a *rigorous* reproducibility study of InPars (Bonifacio et al., 2022). In that, we use *open-source* and *community-trained* generative LLMs (Scao et al., 2022; Wang & Komatsuzaki, 2021), train rankers using multiple seeds, and use statistical testing when measuring improvements. Because efficiency is an important consideration, we also evaluate much smaller ranking models (see Figure 1 and Table 1) compared to those used by Bonifacio et al. (2022).

More specifically, we ask the following research questions:

- **RQ1**: Can we reproduce key findings of InPars (Bonifacio et al., 2022) using *open-source* and *community-trained* LLMs as well as smaller ranking models?

- **RQ2**: Are open-source models more or less useful for generation of synthetic IR training data compared to the similar-sized GPT-3 Curie model (Brown et al., 2020)?

- **RQ3**: Does consistency checking proposed by Dai et al. (2022) improve the InPars recipe? Is it applicable in the purely re-ranking setting as opposed to the retrieval setting (as it was done by Dai et al. (2022))?

- **RQ4**: Can we match performance of large monoT5 rankers—used by Bonifacio et al. (2022)—with much smaller bi-directional Transformer (BERT) models (Devlin et al., 2018; Vaswani et al., 2017)?

- **RQ5**: The smaller monoT5 ranker with 220M parameters used by Bonifacio et al. (2022) does not outperform BM25 for three out of five query sets. Thus, just matching monoT5-220M performance is not enough. Can we instead substantially outperform BM25 using a small and fast ranker such as a MiniLM (Wang et al., 2020) BERT ranker with only 30 million parameters?

Our contributions and findings are as follows:

- We reproduced the key finding by Bonifacio et al. (2022): Generation of synthetic in-domain data using an InPars-like recipe permits training strong in-domain rankers using only a three-shot prompt and in-domain documents, which answers **RQ1**. However, without additional effort such as all-domain pre-training and consistency checking, only a sufficiently large ranking model could outperform BM25 on all datasets.

- We found that *open-source* LLMs BLOOM (Scao et al., 2022) and GPT-J (Wang & Komatsuzaki, 2021), which are trained using only next-token prediction (without further fine-tuning), could be prompted to generate effective synthetic queries. Moreover, using a *community-trained* BLOOM model produced comparable or more accurate[2] ranking models compared to using GPT-3 Curie model (Brown et al., 2020), which addresses **RQ2**.

- We confirmed that consistency checking proposed by Dai et al. (2022) does work for re-rankers and always improves outcomes in the *unsupervised* setting, which answers **RQ3**.

- We also discovered that in the *unsupervised* setting, where synthetic queries were generated using a three-shot prompt, we could match or outperform monoT5 rankers using much smaller BERT ranking models (see Figure 1), which answers **RQ4**. More specifically:
  - We can replace an impractical three-billion parameter monoT5-3B (Nogueira et al., 2020) model with a 7x smaller BERT model while obtaining comparable results. The average gain over BM25 (see Table 1) was 1.32 for monoT5-3B vs. 1.3 for DeBERTA-v3-435M (He et al., 2021) (**RQ1**).
  - Unlike Bonifacio et al. (2022) whose monoT5-220M model with 220 million parameters failed to outperform BM25 on three out of five datasets (unless pre-trained on MS MARCO), we show that a much smaller MiniLM-30M model with only 30 million parameters (Wang et al., 2020) can outperform BM25 by 7%-30% in key metrics (nDCG@K and MRR) when trained using only synthetic training data (**RQ1** and **RQ5**).
  - Outperforming BM25 with a small ranking model such as MiniLM-30M was possible by using: (a) a better model to generate synthetic training data (BLOOM instead of GPT-3 Curie), (b) consistency checking (Dai et al., 2022) (**RQ3**), and (c) all-domain pre-training, each of which helped improve outcomes.

- Obtaining good results in the unsupervised setting described above required re-ranking only 100 candidate documents compared to 1000 used by Bonifacio et al. (2022). Overall, compared to InPars, our training recipe—which we call *InPars-light*—is substantially more cost effective in terms of both, generation of synthetic training data and training/application of ranking models (see § A.2 for a detailed discussion).

---

[2]The only exception was BEIR NQ, where BLOOM-based ranker was 1.4% worse, see Table 4.

- However, when pretraining on MS MARCO was used, the monoT5-220M model was still substantially more accurate than a 7x smaller MiniLM-30M ranker. Moreover, this gap was not reduced by subsequent unsupervised fine-tuning of MiniLM-30M using synthetically generated data. The average gain over BM25 (see Table 1) was 1.46 for monoT5-200M pre-trained on MS MARCO vs. 1.24 for MiniLM-30M pre-trained on MS MARCO and fine-tuned using synthetic training data.

Our code and data are publicly available.[3]

## 2 Related Work

Prompting methods have gained quite a bit of popularity in NLP (see, e.g., Liu et al. (2021) for a recent survey). In particular, prior to the InPars study by Bonifacio et al. (2022), Schick & Schütze (2021) proposed to generate synthetic training sets using in-domain data and zero-shot prompting of LLMs.

However, until recently zero-shot and few-shot prompting of LLMs was not applied to ad hoc retrieval: We know only a few papers directly related to our work. Sachan et al. (2022) were probably the first to demonstrate effectiveness of LLMs in the document ranking task. In their approach—named UPR—they concatenate a document, a special prompt such as "please write a question for this document" and the query itself. Then, UPR uses a pre-trained LLM model to compute the likelihood of generating the query given the passage text. Unlike InPars, they do not use LLM to generate synthetic training data.

Sachan et al. (2022) evaluated their method using only QA (but not IR) datasets and their main results are for an *impractically large* three-billion parameter *instruction-finetuned* model, which was used essentially as a re-ranker (in a zero-shot scenario). The smallest model used by Sachan et al. (2022) had 250 million parameters (compared to our 30-million MiniLM model). It was evaluated *only* on the Natural Questions (NQ) collection (Kwiatkowski et al., 2019) where it outperformed BM25 by about 10%. Although not directly comparable due to using different versions of NQ and model sizes, our $2\times$ larger DeBERTA-v3-435M model outperformed BM25 by 40% while our much smaller MiniLM-30M model with 30 million parameters outperformed BM25 by 15%.

Bonifacio et al. (2022) proposed an *InPars* method, which relied on few-shot prompting. The study had a convincing evaluation on five datasets where only one dataset, namely NQ (Kwiatkowski et al., 2019), was a typical QA collection. Unlike Sachan et al. (2022), Bonifacio et al. (2022) used few-shot prompting to generate synthetic training data for a smaller ranker. For each collection Bonifacio et al. (2022) generated 100K synthetic queries and retained only 10K with the highest average log-probabilities. This can be seen as distillation of an LLM into the ranker.

However, Bonifacio et al. (2022) obtained good results only for a huge monoT5-3B parameter model. They also employed a proprietary GPT-3 model, which can be quite costly to use. In a follow-up study, which is concurrent with this work, Jeronymo et al. (2023) introduced a modification of InPars—dubbed InPars v2— where GPT-3 Curie (Brown et al., 2020) was replaced with an open-source model GPT-J model (Wang & Komatsuzaki, 2021). However, this model swap was "entangled" with at least two other modifications in the training recipe:

- A new query filtering condition that relied on an MS MARCO trained monoT5-3B model.

- The vanilla prompt (which was used in InPars and our experiments) was replaced with the "Guided by Bad Question prompt" (introduced by Bonifacio et al. 2022).

Thus, it is not possible to fairly assess the impact of replacing GPT-3 Curie with GPT-J (Wang & Komatsuzaki, 2021).

An important disadvantage of the InPars v2 recipe is that it is still not cost-effective as authors use a huge monoT5-3B model. The filtering check uses an *expensive* monoT5-3B model trained on MS MARCO

---

[3]https://github.com/searchivarius/inpars_light/

corpus, which is also not always possible in a commercial setting due to licensing issues (MS MARCO is a research-only collection).

Moreover, the monoT5-3B model trained on MS MARCO—albeit being impractical—has excellent zero-shot transferability: Fine-tuning monoT5-3B model trained on MS MARCO with InPars v2 only improves the average BEIR score only by 2.4%: from 0.538 to 0.551. This further complicates assessment of effectiveness of GPT-J.

Dai et al. (2022) used an InPars-like method called Promptagator and created synthetic training data using a huge proprietary FLAN-137B model with 137 billion parameters. Although they used modestly sized retrieval and ranking models with 110 million parameters, Dai et al. (2022) generated as many as million synthetic training queries for each dataset. In contrast, both InPars and InPars-light used only 100K queries per dataset, which was much less expensive (see a discussion in § A.2).

Importantly, Dai et al. (2022) proposed to use consistency checking (Alberti et al., 2019) to filter-out potentially spurious queries, which was not previously done in the IR context. They do not compare with InPars under the same conditions and it was not known if consistency checking would improve the original InPars recipe.

In addition to prompt-based generation of training data, there are multiple proposals for self-supervised adaptation of out-of-domain models using generative pseudo-labeling (Li & Gaussier, 2022; Wang et al., 2022; Reddy et al., 2021). To this end, questions or queries are generated using a pretrained seq2seq model (though an LLMs can be used as well) and negative examples are mined using either BM25 or an out-of-domain retriever or ranker. Unsupervised domain adaptation is complementary to the approaches considered in this work.

The disadvantage of such approaches is that they may need a reasonably effective an out-of-domain ranking model. However, such models can be hard to obtain due to licensing issues and poor transferability from other domains. For example, MS MARCO models have reasonable transferability (Thakur et al., 2021; Mokrii et al., 2021), but MS MARCO cannot be used to train models in a commercial context (without extra licensing from Microsoft). In contrast, the Natural Questions (NQ) collection (Kwiatkowski et al., 2019) has a permissive license[4], but models trained on NQ can fail to transfer to datasets that are not based on Wikipedia (Mokrii et al., 2021).

Another potentially complementary approach is an LLM-assisted query expansion. In particular Gao et al. (2022) prompted a 175B InstructGPT model to generate a hypothetical answer to a question. Then this answer was encoded as a vector and together with the encoding of the original question they were compared with encoded documents. In a purely unsupervised setting—using the Contriever bi-encoder training without supervision (Izacard et al., 2021)—they were able to outperform BM25 by as much as 20%.

Despite strong results, a serious disadvantage of this approach is its dependence on the *external proprietary* model that is costly and inefficient. Although we could not find any reliable benchmarks, a folklore opinion is that GPT generation latency is a few seconds. To verify this, we used the OpenAI playground[5] to generate a few hypothetical answers using the prompt in Gao et al. Gao et al. (2022) and a sample of TREC DL 2020 queries. With a maximum generation length of 256 tokens (a default setting), the latency exceeded four seconds.

Quite interestingly, Gao et al. Gao et al. (2022) tried to replace a 175B GPT-3 model with smaller open-source models on TREC DL 2019 and TREC DL 2020 (see Tables 4 and Table 1 in their study), but failed to obtain consistent and substantial gains over BM25 with models having fewer than 50B parameters.

---

[4]https://github.com/google-research-datasets/natural-questions/blob/master/LICENSE
[5]https://beta.openai.com/playground

Table 2: The format of the vanilla three-shot InPars prompt (Bonifacio et al., 2022)

**Example 1**:
**Document**: <text of the first **example** document>
**Relevant Query**: <text of the first relevant query>

**Example 2:**
**Document:** <text of the second **example** document>
**Relevant Query:** <text of the second relevant query>

**Example 3:**
**Document:** <text of the third **example** document>
**Relevant Query:** <text of the third relevant query>

**Example 4:**
**Document:** <**real in-domain** document text **placeholder**>
**Relevant Query:**

**Notes:** To generate a synthetic query, we first insert a text of a chosen real in-domain document after the prefix "Document:" in example four. Then, we "ask" an LLM to generate a completion.

## 3 Methods

### 3.1 Information Retrieval Pipeline

We use a variant of a classic filter-and-refine multi-stage retrieval pipeline (Matveeva et al., 2006; Prager, 2006; Wang et al., 2011), where top-$k$ candidate documents retrieved by a fast BM25 retriever/scorer (Robertson, 2004) are further re-ranked using a slower neural re-ranker. For collections where documents have titles (NQ BEIR and TREC COVID BEIR), the BM25 retriever itself has two stages: In the first stage we retrieve 1K documents using a Lucene index built over a title concatenated with a main text. In the second stage, these candidates are re-ranked using equally weighted BM25 scores computed separately for the title and the main text.

Our neural rankers are cross-encoder models (Nogueira & Cho, 2019; Lin et al., 2021b), which operate on queries concatenated with documents. Concatenated texts are passed through a backbone bi-directional *encoder-only* Transformer model (Devlin et al., 2018) equipped with an additional ranking head (a fully-connected layer), which produces a relevance score (using the last-layer contextualized embedding of a CLS-token (Nogueira & Cho, 2019)). In contrast, authors of InPars (Bonifacio et al., 2022) use a T5 (Raffel et al., 2020) cross-encoding re-ranker (Nogueira et al., 2020), which is a *full* Transformer model (Vaswani et al., 2017). It uses both the encoder and the decoder. The T5 ranking Transformer is trained to generate labels "true" and "false", which represent relevant and non-relevant document-query pairs, respectively.

Backbone Transformer models can differ in the number of parameters and pre-training approaches (including pre-training datasets). In this paper we evaluated the following models, all of which were pre-trained in the self-supervised fashion without using supervised IR data:

- A six-layer MiniLM-L6 model (Wang et al., 2020). It is a tiny (by modern standards) 30-million parameter model, which was distilled (Li et al., 2014; Romero et al., 2015; Hinton et al., 2015) from Roberta (Liu et al., 2019). We download model L6xH384 MiniLMv2 from the Microsoft website.[6]

- A 24-layer (large) ERNIE v2 model from the HuggingFace hub (Sun et al., 2020)[7]. It has 335 million parameters.

- A 24-layer (large) DeBERTA v3 model with 435 million parameters (He et al., 2021) from the HuggingFace hub [8].

---

[6]`https://github.com/microsoft/unilm/tree/master/minilm`
[7]`https://huggingface.co/nghuyong/ernie-2.0-large-en`
[8]`https://huggingface.co/microsoft/deberta-v3-large`

We chose ERNIE v2 and DeBERTA v3 due to their strong performance on the MS MARCO dataset where they outperformed BERT large (Devlin et al., 2018) and several other models that we tested in the past. Both models performed comparably well in the preliminary experiments, but we chose DeBERTA for main experiments because it was more effective on MS MARCO and TREC-DL 2020. In the post hoc ablation study, DeBERTA outperformed ERNIE v2 on four collections out of five (see Table 4).

However, both of these models are quite large and we aspired to show that an InPars-like training recipe can be used with smaller models too. In contrast, Bonifacio et al. (2022) were able to show that *only* a big monoT5-3B model with 3B parameters could outperform BM25 on all five datasets: The smaller monoT5-200M ranker with 200 million parameters, which is still quite large, outperformed BM25 only on MS MARCO and TREC-DL 2020.

## 3.2 Generation of Synthetic Training Data

We generate synthetic training data using a well-known few-shot prompting approach introduced by Brown et al. (2020). In the IR domain, it was first used by Bonifacio et al. (2022) who called it *InPars*. The key idea of InPars is to "prompt" a large language model with a few-shot textual demonstration of known relevant query-document pairs. To produce a novel query-document pair, Bonifacio et al. (2022) appended an in-domain document to the prompt and "asked" the model to complete the text. Bonifacio et al. (2022) evaluated two types of the prompts of which we use only the so-called vanilla prompt (see Table 2).

As in the InPars study (Bonifacio et al., 2022), we generated 100K queries for each dataset with exception of MS MARCO and TREC DL.[9] Repeating this procedure for many in-domain documents produces a large training set, but it can be quite imperfect. In particular, we carried out spot-checking and found quite a few queries that were spurious or only tangentially relevant to the passage from which they were generated.

Many spurious queries can be filtered out automatically. To this end, Bonifacio et al. (2022) used only 10% of the queries with the highest log-probabilities (averaged over query tokens). In the Promptagator recipe, Dai et al. (2022) used a different filtering procedure, which was a variant of consistency checking (Alberti et al., 2019). Dai et al. (2022) first trained a retriever model using all the generated queries. Using this retriever, they produced a ranked set of documents for each query. The query passed the consistency check if the first retrieved document was the document from which the query was generated. A straightforward modification of this approach is to check if a generated document is present in a top-$k$ ($k > 1$) candidate set produced by the retriever. Dai et al. (2022) used consistency checking with bi-encoding retrieval models, but it is applicable to cross-encoding re-ranking models as well.

## 3.3 InPars-light Training Recipe

The InPars-light is not a new method. It is a training recipe, which a modification of the original InPars. Yet, it is substantially more cost effective for generation of synthetic queries, training the models, and inference. InPars-light has the following main "ingredients":

- Using open-source models instead of GPT-3;

- Using smaller ranking BERT models instead of monoT5 rankers;

- Fine-tuning models on consistency-checked training data;

- Optional pre-training of models using *all* generated queries from *all* collections.

- Re-ranking only 100 candidate documents instead of 1000: However, **importantly**, the training procedure still generates negatives from a top-1000 set produced by a BM25 ranker.

To obtain consistency-checked queries for a given dataset, a model trained on InPars-generated queries (for this dataset) was used to re-rank output of *all* original queries (for a given dataset). Then, all the queries

---

[9]Because both datasets use the same set of passages they share the same set of 100K generated queries.

where the query-generating-document did *not* appear among top-$k$ scored documents were discarded. In our study, we experimented with $k$ from one to three (but *only* on MS MARCO).[10] Although $k = 1$ worked pretty well, using $k = 3$ lead to a small boost in accuracy. Consistency-checking was carried out using DeBERTA-v3-435M (He et al., 2021). We want to emphasize that consistency-checked training data was used *in addition* to original InPars-generated data (but not instead), namely, to fine-tune a model initially trained on InPars generated data.

Also, quite interestingly, a set of consistency-checked queries had only a small (about 20-30%) overlap with the set of queries that were selected using the original InPars recipe (based on average log-probabilities). Thus, consistency-checking increased the amount of available training data. It might seem appealing to achieve the same objective by simply picking a larger number of queries (with highest average log-probabilities). However, preliminary experiments on MS MARCO showed that a naive increase of the number of queries degraded effectiveness (which is consistent with findings by Bonifacio et al. (2022)).

Although, the original InPars recipe with open-source models and consistency checking allowed us to train strong DeBERTA-v3-435M models, performance of MiniLM models was lackluster (roughly at BM25 level for all collections).

Because bigger models performed quite well, it may be possible to distill (Li et al., 2014; Romero et al., 2015; Hinton et al., 2015) their parameters into a much smaller MiniLM-30M model. Distillation is known to be successful in the IR domain (Hofstätter et al., 2020; Lin et al., 2020), but it failed in our case. Thus we used the following workaround instead:

- First we carried out an *all-domain* pre-training *without any filtering* (i.e., using *all* queries from *all* collections);

- Then, we fine-tuned all-domain pre-trained models on the consistency-checked in-domain data for each collection separately.

### 3.4 Miscellaneous

We carried out experiments using FlexNeuART Boytsov & Nyberg (2020), which provided support for basic indexing, retrieval, and neural ranking. Both generative and ranking models were implemented using PyTorch and Huggingface (Wolf et al., 2020). Ranking models were trained using the InfoNCE loss (Le-Khac et al., 2020). In a single training epoch, we selected randomly one pair of positive and three negative examples per query (negatives were sampled from 1000 documents with highest BM25 scores). Note that, however, that during inference we re-ranked only 100 documents. In preliminary experiments on MS MARCO we used to sample from a top-100 set as well. However, the results were surprisingly poor and we switched to sampling from a top-1000 set (we did not try any other sampling options though). A number of negatives was not tuned: We used as much as we can while ensuring we do not run out of GPU memory during training on any collection.

We used the AdamW optimizer (Loshchilov & Hutter, 2017) with a small weight decay ($10^{-7}$), a warm-up schedule, and a batch size of 16.[11] We used different base rates for the fully-connected prediction head ($2 \cdot 10^{-4}$) and for the main Transformer layers ($2 \cdot 10^{-5}$). The mini-batch size was equal to one and a larger batch size was simulated using a 16-step gradient accumulation. We did not tune optimization parameters and chose the values based on our prior experience of training neural rankers for MS MARCO.

We trained each ranking model using three seeds and reported the *average results* (except for the best-seed analysis in Table 5). Statistical significance is computed between "seed-average" runs where query-specific metric values are first averaged over all seeds and then a standard paired difference test is carried out using these seed-average values (see § A.1 for details).

---

[10]We did not want to optimize this parameter for all collections and, thus, to commit a sin of tuning hyper-parameters on the complete test set.

[11]The learning rate grows linearly from zero for 20% of the steps until it reaches the base learning rate (Mosbach et al., 2020; Smith, 2017) and then goes back to zero (also linearly).

Except zero-shot experiments, we trained a separate model for each dataset, which is consistent with Bonifacio et al. (2022). Moreover, we computed exactly the same accuracy metrics as Bonifacio et al. (2022). For statistical significance testing we used a paired two-sided t-test. For query sets with a large number of queries (MS MARCO development set and BEIR Natural Questions) we used a lower threshold of 0.01. For small query sets (Robust04, TREC DL, and TREC-COVID), the statistical significance threshold was set to 0.05.

We implemented our query generation module using the AutoModelForCasualLM interface from HuggingFace. We used a three-shot vanilla prompt template created by Bonifacio et al. (2022) (also shown in Table 2). The output was generated via greedy decoding. The maximum number of new tokens generated for each example was set to 32. Note that query generation was a time-consuming process even though we used open-source models. Thus, we did it only once per dataset, i.e., without using multiple seeds.

## 4 Datasets

Because we aimed to reproduce the main results of InPars (Bonifacio et al., 2022), we used exactly the same set of queries and datasets, which are described below. Except MS MARCO (which was processed directly using FlexNeuART Boytsov & Nyberg (2020) scripts), datasets were ingested with a help of the IR datasets package (MacAvaney et al., 2021).

Some of the collections below have multiple text fields, which were used differently between BM25 and neural ranker. All collections except Robust04 have exactly one query field. Robust04 queries have the following parts: title, description, and narrative. For the purpose of BM25 retrieval and ranking, we used only the title field, but the neural ranker used only the description field (which is consistent with Bonifacio et al. 2022). The narrative field was not used.

Two collections have documents with both the title and the main body text fields (NQ BEIR and TREC COVID BEIR). The neural rankers operated on concatenation of these fields. If this concatenation was longer than 477 BERT tokens, the text was truncated on the right (queries longer than 32 BERT tokens were truncated as well). For BM25 scoring, we indexed concatenated fields as well in Lucene. However, after retrieving 1000 candidates, we re-ranked them using the sum of BM25 scores computed separately for the title and the main body text fields (using FlexNeuART Boytsov & Nyberg (2020)).

**Synthetically Generated Training Queries.** For each of the datasets, Bonifacio et al. (2022) provided both the GPT-3-generated queries (using GPT-3 Curie model) and the documents that were used to generate the queries. This permits a fair comparison of the quality of training data generated using GPT-3 Curie with the quality of synthetic training data generated using open-source models GPT-J (Wang & Komatsuzaki, 2021) and BLOOM (Scao et al., 2022). According to the estimates of Bonifacio et al. (2022), the Curie model has 6B parameters, which is close to the estimate made by by Gao from EleutherAI Gao (2021). Thus, we used GPT-J (Wang & Komatsuzaki, 2021) and BLOOM (Scao et al., 2022) models with 6 and 7 billion parameters, respectively. Although other open-source models can potentially be used, generation of synthetic queries is quite expensive and exploring other open-source options is left for future work.

**MS MARCO sparse and TREC DL 2020.** MS MARCO is collection of 8.8M passages extracted from approximately 3.6M Web documents, which was derived from the MS MARCO reading comprehension dataset (Bajaj et al., 2016; Craswell et al., 2020). It "ships" with more than half a million of question-like queries sampled from the Bing search engine log with subsequent filtering. The queries are not necessarily proper English questions, e.g., "lyme disease symptoms mood", but they are answerable by a short passage retrieved from a set of about 3.6M Web documents (Bajaj et al., 2016). Relevance judgements are quite sparse (about one relevant passage per query) and a positive label indicates that the passage can answer the respective question.

The MS MARCO collections has several development and test query sets of which we use only a development set with approximately 6.9K sparsely-judged queries and the TREC DL 2020 (Craswell et al., 2020) collection of 54 densely judged queries. Henceforth, for simplicity when we discuss the MS MARCO development set we use a shortened name MS MARCO, which is also consistent with Bonifacio et al. (2022).

Note that the MS MARCO collection has a large training set, but we do not use it in the fully unsupervised scenario. It is used only supervised transfer learning (see § 5).

**Robust04** (Voorhees, 2004) is a small (but commonly used) collection that has about 500K news wire documents. It comes with a small but densely judged set of 250 queries, which have about 1.2K judgements on average.

**Natural Questions (NQ) BEIR** (Kwiatkowski et al., 2019) is an open domain Wikipedia-based Question Answering (QA) dataset. Similar to MS MARCO, it has real user queries (submitted to Google). We use a BEIR's variant of NQ (Thakur et al., 2021), which has about 2.6M short passages from Wikipedia and 3.4K sparsely-judged queries (about 1.2 relevant documents per query).

**TREC COVID BEIR** (Roberts et al., 2020) is a small corpus that has 171K scientific articles on the topic of COVID-19 and. TREC COVID BEIR comes with 50 densely-judged queries (1.3K judged documents per query on average). It was created for a NIST challenge whose objective was to develop information retrieval methods tailored for the COVID-19 domain (with a hope to be a useful tool during COVID-19 pandemic). We use the BEIR's version of this dataset (Thakur et al., 2021).

## 5    Results

The summary of experimental results is provided in Figure 1 and Table 1. Our detailed experimental results are presented in Table 3. Note that in addition to our own measurements, we copy key results from prior work (Nogueira et al., 2020; Bonifacio et al., 2022), which include results for BM25 (by Bonifacio et al. (2022)), re-ranking using OpenAI API, and monoT5 rankers. In our experiments, we statistically test several hypotheses, which are explained separately at the bottom of each table.

**BM25 baselines.** To assess the statistical significance of the difference between BM25 and a neural ranker, we had to use our own BM25 runs. These runs were produced using FlexNeuART Boytsov & Nyberg (2020). Comparing effectiveness of FlexNeuART Boytsov & Nyberg (2020) BM25 with effectiveness of Pyserini (Lin et al., 2021a) BM25—used the InPars study (Bonifacio et al., 2022)—we can see that on all datasets except TREC DL 2020 we closely match (within 1.5%) Pyserini numbers. On TREC DL 2020 our BM25 is 6% more effective in nNDCG@10 and 25% more effective in MAP.

**Unsupervised-only training (using three-shot prompts).** We consider the scenario where synthetic training data is generated using a three-shot prompt to be unsupervised. Although the prompt is based on human supervision data (three random samples from the MS MARCO training corpus), these samples are not directly used for training, but only to generate synthetic data.

In this scenario, we reproduce the key finding by Bonifacio et al. (2022): Generation of synthetic in-domain data using an InPars-like recipe permits training strong in-domain rankers using only a three-shot prompt and in-domain documents. However, if we use the original InPars recipe, only a large ranking model (DeBERTA-v3-435M) consistently outperforms BM25. This answers **RQ1**. With DeBERTA-v3-435M we obtain accuracy similar to that of monoT5-3B on four collections out of five, even though monoT5-3B has 7x more parameters. The average gain over BM25 is 1.3 (for DeBERTA-v3-435M) vs 1.32 for monoT5-3B (see Table 1).

Accuracy of our smallest model MiniLM-L6-30M with all-domain pretraining and finetuning on consistency-checked data (referred to as InPars **all ▶** consist. check in Table 3) roughly matches that of the 7x larger monoT5-220M on MS MARCO and TREC DL 2020. Yet, it is substantially better than monoT5-220M on the remaining datasets, where monoT5-220M effectiveness is largely at BM25 level: The average gain over BM25 (see Table 1) is 1.07 for monoT5-200M vs. 1.13 for MiniLM-30M. MiniLM-L6-30M outperforms BM25 on all collections and all metrics. In all but one case these differences are also *statistically significant*. In terms of nDCG and/or MRR, MiniLM-30M is 7%-30% more accurate than BM25.

In summary, we can replace monoT5 rankers with much smaller BERT models while obtaining comparable or better average gains over BM25. This answers **RQ4**.

**Impact of consistency checking and all-domain pre-training.** We found that, on its own, the InPars recipe did not produce a strong MiniLM-L6-30M ranking model. This is in line with the findings of Bonifacio

Table 3: Model Accuracy for Various Scenarios (averaged over three seeds)

| | MS MARCO | TREC DL 2020 | | Robust04 | | NQ | TREC COVID |
|---|---|---|---|---|---|---|---|
| | MRR | MAP | nDCG@10 | MAP | nDCG@20 | nDCG@10 | nDCG@10 |
| BM25 (Bonifacio et al., 2022) | 0.1874 | 0.2876 | 0.4876 | 0.2531 | 0.4240 | 0.3290 | 0.6880 |
| BM25 (this study) | 0.1867 | 0.3612 | 0.5159 | 0.2555 | 0.4285 | 0.3248 | 0.6767 |
| **OpenAI Ranking API**: re-ranking 100 Documents (Bonifacio et al., 2022) | | | | | | | |
| Curie (6B) (Bonifacio et al., 2022) | $ | 0.3296 | 0.5422 | 0.2785 | 0.5053 | 0.4171 | 0.7251 |
| Davinci (175B) (Bonifacio et al., 2022) | $ | 0.3163 | 0.5366 | 0.2790 | 0.5103 | $ | 0.6918 |
| **Unsupervised**: InPars-based Training Data (three-shot prompting) | | | | | | | |
| monoT5-220M (InPars) (Bonifacio et al., 2022) | 0.2585 | 0.3599 | 0.5764 | 0.2490 | 0.4268 | 0.3354 | 0.6666 |
| monoT5-3B (InPars) (Bonifacio et al., 2022) | **0.2967** | 0.4334 | 0.6612 | **0.3180** | 0.5181 | **0.5133** | 0.7835 |
| MiniLM-L6-30M (InPars) | [ba]0.2117 | [b]0.3482 | [b]0.4953 | [ba]0.2263 | [ba]0.3802 | [ba]0.2187 | [b]0.6361 |
| MiniLM-L6-30M (InPars ▶ consist. check) | [cba]0.2336 | [cb]0.3769 | [cb]0.5543 | [cb]0.2556 | [cb]0.4440 | [cb]0.3239 | [cb]0.6926 |
| MiniLM-L6-30M (InPars **all** ▶ consist. check) | [ca]0.2468 | [ca]0.3929 | [ca]0.5726 | [c]0.2639 | [ca]0.4599 | [ca]0.3747 | [ca]0.7688 |
| DeBERTA-v3-435M (InPars) | [ba]0.2746 | [ba]0.4385 | [a]0.6649 | [ba]0.2811 | [ba]0.4987 | [ba]0.4476 | [a]0.8022 |
| DeBERTA-v3-435M (InPars ▶ consist. check) | [cba]0.2815 | [cba]**0.4446** | [ca]**0.6717** | [cba]0.3009 | [cba]**0.5360** | [cba]0.4621 | [ca]**0.8183** |
| DeBERTA-v3-435M (InPars **all** ▶ consist. check) | [ca]0.1957 | [c]0.3607 | [c]0.5007 | [c]0.2518 | [c]0.4320 | [c]0.3267 | [c]0.6953 |

**Supervised transfer learning with optional unsupervised fine-tuning**: transfer from MS MARCO with optional fine-tuning on consistency-checked InPars data

| | MS MARCO | TREC DL 2020 | | Robust04 | | NQ | TREC COVID |
|---|---|---|---|---|---|---|---|
| | MRR | MAP | nDCG@10 | MAP | nDCG@20 | nDCG@10 | nDCG@10 |
| MiniLM-L6-30M (MS MARCO) | [da]0.3080 | [a]0.4370 | [a]0.6662 | [da]0.2295 | [da]0.3923 | [da]0.4646 | [da]0.7476 |
| MiniLM-L6-30M (MS MARCO ▶ consist. check) | [da]0.2944 | [a]0.4311 | [a]0.6501 | [da]0.2692 | [da]0.4730 | [da]0.4320 | [da]0.7898 |
| DeBERTA-v3-435M (MS MARCO) | [da]0.3508 | [a]0.4679 | [da]0.7269 | [a]0.2986 | [a]0.5304 | [da]0.5616 | [a]0.8304 |
| DeBERTA-v3-435M (MS MARCO ▶ consist. check) | [da]0.3166 | [a]0.4553 | [da]0.6912 | [a]0.3011 | [a]0.5371 | [da]0.5075 | [a]0.8165 |
| monoT5-220M (MS MARCO) (Nogueira et al., 2020) | 0.3810 | 0.4909 | 0.7141 | 0.3279 | 0.5298 | 0.5674 | 0.7775 |
| monoT5-3B (MS MARCO) (Nogueira et al., 2020) | **0.3980** | **0.5281** | **0.7508** | 0.3876 | 0.6091 | **0.6334** | 0.7948 |
| monoT5-3B (MS MARCO ▶ InPars) (Bonifacio et al., 2022) | 0.3894 | 0.5087 | 0.7439 | **0.3967** | **0.6227** | 0.6297 | **0.8471** |

OpenAI API ranking results were produced by Bonifacio et al. (2022): $ denotes experiments that were too expensive to run.
*InPars* denotes the original query-generation method with filtering-out 90% of queries having lowest average log-probabilities.
*InPars all* denotes the query-generation method without query filtering, which was used in *all-domain* pretraining.
*Consist. checked queries* denotes a set of generated queries filtered out (via consistency checking) using the **DeBERTA-v3-435M** model trained on InPars-generated data.

Best results are marked by **bold** font **separately** for each training scenario.

Super-scripted labels denote the following statistically significant differences (thresholds are given in the main text):
**a**: between a given neural ranking model and BM25;
**b**: between (InPars) and (InPars ▶ consist. check) when comparing neural ranking models of **same** type.
**c**: between (InPars **all** ▶ consist. check) and (InPars ▶ consist. check) when comparing neural ranking models of **same** type.
**d**: between (MS MARCO) and (MS MARCO ▶ consist. check) when comparing neural ranking models of **same** type.

Table 4: Performance of InPars for Different Generating and Ranking Models.

| | MS MARCO MRR | TREC DL 2020 MAP | nDCG@10 | Robust04 MAP | nDCG@20 | NQ nDCG@10 | TREC COVID nDCG@10 |
|---|---|---|---|---|---|---|---|
| BM25 (ours) | 0.1867 | 0.3612 | 0.5159 | 0.2555 | 0.4285 | 0.3248 | 0.6767 |
| ERNIE-v2-335M GPT-3 Curie (6B) | [a]0.2538 | [a]0.4140 | [a]0.6229 | [a]0.2357 | 0.4016 | [a]0.4277 | [a]0.7411 |
| ERNIE-v2-335M GPT-J (6B) | [ba]0.2608 | [ba]0.4286 | [a]0.6367 | [cb]0.2691 | [cba]0.4724 | [a]0.4248 | [ba]0.7750 |
| ERNIE-v2-335M BLOOM (7B) | [dba]0.2605 | [ba]0.4286 | [a]0.6407 | [cba]**0.2852** | [dcba]**0.5102** | [da]0.4215 | [ba]0.7871 |
| DeBERTA-v3-435M BLOOM (7B) | [dba]**0.2746** | [ba]**0.4385** | [ba]**0.6649** | [ba]0.2811 | [dba]0.4987 | [dba]**0.4476** | [ba]**0.8022** |

**Notes:** Best results are in bold. Super-scripted labels denote statistically significant differences (thresholds are given in the main text):
**a**: between a given neural ranking model and BM25;
**b**: between a given neural ranking model and ERNIE-v2-335M trained using OpenAI GPT-3 Curie.
**c**: between two ERNIE models trained using GPT-J-generated queries and BLOOM-generated queries;
**d**: between the DeBERTA model and the ERNIE model trained using BLOOM-generated queries.

et al. (2022), who observed that only monoT5-3B (but not a much smaller monoT5-220M) outperformed BM25 on all collections. Strong performance of MiniLM-L6-30M in our study was due to additional training with consistency-checked data and pre-training on all-domain data (all queries from all collections). To confirm the effectiveness of these procedures, we carried out ablation experiments.

Recall that the consistency-checked training data was produced using only the DeBERTA-v3-435M model. Moreover, this data was used only to fine-tune a model that was pre-trained using data generated by the original InPars recipe. From Table 3, we can see that for both MiniLM-L6-30M and DeBERTA-v3-435M fine-tunining on consistency-checked data improves outcomes (which answers **RQ3**): For 12 measurements out of 14, these improvements are statistically significant (denoted by super-script label "b").

Moreover, all-domain pretraining (instead of training on data generated by the original InPars recipe) further boosts accuracy of MiniLM-L6-30M in all cases: All these improvements are statistically significant (denoted by super-script label "c"). In contrast, all-domain pretraining substantially degrades performance of DeBERTA-v3-435M. An in-depth investigation showed that for one seed (out of three), the model has failed to converge properly. Therefore, we also analyze the best-seed outcomes which are presented in § A.3 Table 5. For MiniLM-L6-30M, the all-domain pre-training improves the best-seed accuracy in all cases. For DeBERTA-v3-435M, there is either a substantial degradation or a small decrease/increase that is not statistically significant (denoted by super-script label "c"). Thus, our biggest model—unlike a 15x smaller MiniLM-L6-30M—does not benefit from all-domain pretraining. However, there is no substantial degradation either.

**Supervised transfer learning with optional unsupervised fine-tuning.** We found that our ranking models trained on MS MARCO (both MiniLM-L6-30M and DeBERTA-v3-435M) transferred well to other collections in almost all the cases. However, monoT5 models trained on MS MARCO are still substantially more accurate. According to Table 1, the average gains over BM25 are (1) 1.21 for MiniLM-30M vs. 1.46 for monoT5-200M and (2) 1.42 for DeBERTA-v3-435M vs. 1.59 for monoT5-3B. In that, this gap is not reduced by fine-tuning using synthetically generated data. This is different from the fully unsupervised scenario described above, where MiniLM-L6-30M often outperforms monoT5-220M while DeBERTA-v3-435M is at par with monoT5-3B.

This is in line with prior findings that large ranking models have better zero-shot transferring effectiveness (Ni et al., 2021; Rosa et al., 2022). However, using multi-billion parameter models pre-trained on MS MARCO in a commercial setting is problematic from both efficiency and legal standpoints. In particular, MS MARCO has a research-only license.[12].

**Model-type ablation.** To assess the impact of replacing GPT-3 Curie with an open-source model, we carried out experiments using the following ranking models: ERNIE-v2 (Sun et al., 2020) and DeBERTA-v3-435M (He et al., 2021). According to Table 4, except for NQ—where all generative models were equally good—both

---

[12]See terms and conditions: `https://microsoft.github.io/msmarco/`

GPT-J (Wang & Komatsuzaki, 2021) and BLOOM (Scao et al., 2022) outperformed GPT-3 Curie. This answers **RQ2**.

The difference in accuracy was particularly big for Robust04. The average relative gain over GPT-3 curie (not shown in the table) were 7.2% for BLOOM and 5.2% for GPT-J.[13] Out of 14 comparisons, 10 were statistically significant (as denoted by super-script "b").

In addition to varying a generative model, we assessed the impact of using DeBERTA-v3 instead of ERNIE-v2. This time around, both models were trained using BLOOM-generated queries. We can see that DeBERTA-v3 was better than ERNIE-v2 except the case of Robust04.

## 6 Conclusion

We carried out a reproducibility study of InPars (Bonifacio et al., 2022), which is a method for unsupervised training of neural rankers. As a by-product of this study, we developed a simple-yet-effective modification of InPars, which we called InPars-light. Unlike InPars, InPars-light uses only a community-trained open-source language model BLOOM (with 7B parameters), 7x-100x smaller ranking models, and re-ranks only top-100 candidate records instead of top-1000.

Not only were we able to reproduce key findings from prior work (Bonifacio et al., 2022), but, combining the original InPars recipe (Bonifacio et al., 2022) with (1) fine-tuning on consistency-checked data (Dai et al., 2022) and (2) all-domain pretraining, we trained an efficient yet small model MiniLM-L6-30M consistently outperforming BM25 in the unsupervised setting. In the same scenario, using a larger DeBERTA-v3-435M model, we largely matched performance of a 7x larger monoT5-3B.

In the supervised transfer learning setting—when pretraining on MS MARCO was used—the monoT5-220M model was still substantially more accurate than a 7x smaller MiniLM-30M ranker and this gap was not reduced by unsupervised fine-tuning using synthetically generated data.

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

# A   Appendix

## A.1   Statistical Testing with Multiple-Seed Models

To compute statistical significance using a paired statistical test between results from models $A$ and $B$, one first has to compute the values of an accuracy metric (e.g., MRR) for each query separately. Let $m_i^A$ and $m_i^B$

be sequences of query-specific metric values for models $A$ and $B$, respectively. The paired statistical test is then carried out using a sequence of differences $m_i^A - m_i^B$.

This procedure is not directly applicable when each model is presented by multiple outcomes/seeds. To overcome this issue, we (1) obtain a set of query- and seed-specific metric values, and (2) average them over seeds, thus, reducing the problem to a single-seed statistical testing. In more details, let $m_{is}^A$ and $m_{is}^B$ be sets of query- and seed-specific metric values for models $A$ and $B$, respectively. Recall that we have three seeds, so $s \in \{1, 2, 3\}$. Then, we obtain seed-average runs $m_i^A = 1/3 \sum_{s=1}^{3} m_{is}^A$ and $m_i^B = 1/3 \sum_{s=1}^{3} m_{is}^B$ and compute statistical significance using a paired difference test.

## A.2 Cost and Efficiency

In the following sub-section, we discuss both the ranking efficiency and query-generation cost. Although one may argue that the cost of generation using open-source models is negligibly small, in reality this is true only if one owns their own hardware and generates enough queries to justify the initial investment. Thus, we make a more reasonable assessment assuming that the user can employ a cheap cloud service.

**Cost of Query Generation.** For the original InPars Bonifacio et al. (2022), the cost of generation for the GPT-3 Curie model is $0.002 per one thousand tokens. The token count includes the length of the prompt and the prompting document.[14] We estimate that (depending on the collection) a single generation involves 300 to 500 tokens: long-document collections Robust04 and TREC-COVID both have close to 500 tokens per generation.

Taking an estimate of 500 tokens per generation, the cost of querying OpenAI GPT-3 Curie API can be up to $100 for Robust04 and TREC-COVID. Assuming that sampling from the 137-B FLAN model (used by (Dai et al., 2022)) to be as expensive as from the largest GPT-3 model Davinci (which has a similar number of parameters), each generation in the Promptagator study (Dai et al., 2022), was 10x more expensive compared to InPars study (Bonifacio et al., 2022). Moreover, because Dai et al. (2022) generated one million samples per collection, the Promptagator recipe was about *two orders* of magnitude more expensive compared to InPars.

In contrast, it takes only about 15 hours to generate 100K queries using RTX 3090 GPU. Extrapolating this estimate to A100, which is about 2x faster than RTX 3090[15], and using the pricing of Lambda GPU cloud, we estimate the cost of generation in our InPars-light study to be under $10 per collection. [16]

**Efficiency of Re-ranking.** A rather common opinion (in particular expressed by anonymous reviewers on multiple occasions) is that using cross-encoders is not a practical option. This might be true for extremely constrained latency environments or very large models, but we think it is totally practical to use small models such as MiniLM-L6-30M for applications such as enterprise search. In particular, on a reasonably modern GPU (such as RTX 3090) and MinLm-L6-30M re-ranking throughput exceeds 500 passages per second (assuming truncation to the first 477 characters). Thus re-ranking 100 documents has an acceptable sub-second latency. In fact, Cohere AI provides re-ranking with neural models as a cloud service.[17]

**Cost of Model Training.** Here, all training times are given with respect to a single RTX 3090 GPU. Training and evaluating MiniLM6-30M models had *negligible* costs dominated by all-domain pretraining, which took about two hours per seed. In contrast, the all-domain pretraining of DeBERTA-v3-435M took 28 hours. However, without all-domain pretraining, the training time itself was rather small, in particular, because we used only a fraction of all generated queries (10K queries in the original InPars training and about 20K queries in the follow-up fine-tuning using consistency checked data).

Aside from all-domain pre-training, the two most time-consuming operations were:

- Evaluation of model effectiveness on large query sets MS MARCO and NQ, which jointly have about 10K queries;

---

[14]https://chengh.medium.com/understand-the-pricing-of-gpt3-e646b2d63320
[15]https://lambdalabs.com/blog/nvidia-rtx-a6000-benchmarks
[16]https://lambdalabs.com/service/gpu-cloud#pricing
[17]https://docs.cohere.com/docs/reranking

Table 5: Best-Seed Results for Unsupervised Training

| | MS MARCO MRR | TREC DL 2020 MAP | nDCG@10 | Robust04 MAP | nDCG@20 | NQ nDCG@10 | TREC COVID nDCG@10 |
|---|---|---|---|---|---|---|---|
| BM25 (ours) | 0.1867 | 0.3612 | 0.5159 | 0.2555 | 0.4285 | 0.3248 | 0.6767 |
| **MiniLM-L6-30M** results | | | | | | | |
| MiniLM (InPars) | $^{ba}$0.2197 | $^{b}$0.3562 | $^{b}$0.5151 | $^{ba}$0.2380 | $^{ba}$0.4029 | $^{ba}$0.2415 | $^{b}$0.6732 |
| MiniLM (InPars ▶ consist. check) | $^{cba}$0.2422 | $^{b}$0.3844 | $^{ba}$0.5753 | $^{cb}$0.2615 | $^{cba}$0.4554 | $^{cb}$0.3297 | $^{ba}$0.7483 |
| MiniLM (InPars **all** ▶ consist. check) | $^{ca}$**0.2517** | $^{a}$**0.3945** | $^{a}$**0.5769** | $^{c}$**0.2671** | $^{ca}$**0.4691** | $^{ca}$**0.3800** | $^{a}$**0.7709** |
| **DeBERTA-v3-435M** results | | | | | | | |
| DeBERTA (InPars) | $^{ba}$0.2748 | $^{a}$0.4437 | $^{a}$0.6779 | $^{ba}$0.2874 | $^{ba}$0.5131 | $^{a}$0.4872 | $^{a}$0.8118 |
| DeBERTA (InPars ▶ consist. check) | $^{ba}$**0.2847** | $^{a}$**0.4479** | $^{a}$**0.6813** | $^{ba}$0.3043 | $^{ba}$0.5417 | $^{ca}$**0.4924** | $^{a}$**0.8305** |
| DeBERTA (InPars **all** ▶ consist. check) | $^{a}$0.2804 | $^{a}$0.4414 | $^{a}$0.6575 | $^{a}$**0.3076** | $^{a}$**0.5505** | $^{ca}$0.4746 | $^{a}$0.8259 |

**Notes:** Best results are marked by **bold** font (**separately** for each model).

*Consist. checked queries* denotes a set of generated queries filtered out (via consistency checking) using the **DeBERTA-v3-435M** model trained on InPars-generated data.

Super-scripted labels denote the following statistically significant differences (thresholds are given in the main text):

**a**: between a given neural ranking model and BM25;

**b**: between (InPars) and (InPars ▶ consist. check) when comparing ranking models of **same** type.

**c**: between (InPars **all** ▶ consist. check) and (InPars ▶ consist. check) when comparing ranking models of **same** type.

- Consistency checking using DeBERTA-v3-435M model.

The total effectiveness evaluation time for DeBERTA-v3-435 was about 6 hours (for all collections). The consistency checking, however, took about 48 hours. In the future, we may consider carrying out consistency checking using a much faster model, such as MiniLM-L6-30M.

## A.3    Additional Experimental Results

Our rankers were trained using three seeds. However, in the case of all-domain pretraining, DeBERTA converged poorly for one seed. Therefore, in Table 5 we present best-seed results.

