# OpenReview forum: "InPars-Light: Cost-Effective Unsupervised Training of Efficient Rankers"
_TMLR — Accepted by TMLR_

### Review · Reviewer_KGAf · 2023-08-29

**Summary Of Contributions:**

-- The paper presents a reproduction study of InPars (Bonifacio et al., 2022), a method for unsupervised training of neural rankers using synthetic training data generated via few-shot prompting of large language models (LLMs).

-- As a byproduct of the reproduction study, the authors propose InPars-Light, a more cost-effective modification of InPars that uses smaller ranking models like MiniLM and DeBERTA instead of large monoT5 models, and generates training data using the open-source BLOOM LLM instead of proprietary GPT-3.

--Key findings are:

 - InPars-Light with DeBERTA-v3 (435M params) matches effectiveness of the much larger monoT5-3B (3B params) at lower cost.

 - Even smaller 30M parameter MiniLM models consistently outperform BM25 when trained with InPars-Light.

 - Open-source BLOOM model generates better synthetic training data than GPT-3 for this task.

 - Additional techniques like consistency checking and all-domain pretraining further improve InPars-Light results.

**Audience:**

Yes

**Broader Impact Concerns:**

-- The techniques presented could make prompt-based training of neural rankers more accessible due to reduced computational requirements.

-- Reliance on proprietary models like GPT-3 is reduced through the use of open-source LLMs.

-- No major societal concerns identified with the presented techniques.

**Claims And Evidence:**

Yes

**Requested Changes:**

-- Analyze if there are benefits from using other large open-source LLMs for training data generation.

-- Study the impact of different consistency checking thresholds more thoroughly.

**Strengths And Weaknesses:**

The paper is well-written and puts itself nicely in context of past work.

Strengths:

-- Rigorous experimental methodology. The authors use proper statistical testing to validate effectiveness gains over baselines. This strengthens the reliability of the results.

--Comprehensive evaluation on multiple standard IR datasets. The method is evaluated extensively on 5 datasets from different domains. This demonstrates broad applicability.

-- Detailed cost and efficiency analysis. The paper provides a detailed breakdown of computational costs and latency measurements. This is useful for assessing real-world viability.


Weaknesses:

-- This paper is just a major replication analysis and no new insights are presented. Essentially, the paper provide sa "recipe" for faster InPars but provides no theoretical guarantees of any type. It is just a massive empirical exercise showing several things that "worked."


--Does not explore other open-source LLMs besides BLOOM for data generation. Only BLOOM is compared to GPT-3 for training data generation. Studying other large open-source LLMs could reveal further improvements.

--Limited analysis on the impact of different consistency checking thresholds. Consistency checking is shown to help, but only a single threshold is tried. More analysis on the threshold could be beneficial.

--No exploration of other prompt formats beyond the vanilla 3-shot prompt. The impact of other prompting strategies is not studied. This could potentially further improve results.


-- Provides limited details on all-domain pretraining procedure. More implementation details on this technique that improved MiniLM results could be useful.

---

> ### Author Response · Authors · 2023-08-29
> **author response**
>
> Thank you very much for the thoughtful and quick review!
>
> 1) Please, note that our paper is not a *replication* study, but rather a reproducibility study: It tests if the original method InPars works under **different** conditions (to emphasize we do not replicate original results directly). TMLR welcomes reproducibility (and replicability) studies. Thus, we cannot agree that providing a reproducibility study is a weakness.
>
> 2) We did test GPT-J (see Table 4) **in addition** to BLOOM. Because experimentation is rather massive (multiple collection generation and training rankers using three seeds), it is not feasible to do an extensive testing of OSS models, especially, because it is not **required** to answer our main research question: Does a community-trained model **without** instruction, tuning learns something useful about information retrieval? We know that BERT on its own does not work well without finetuning, but LLMs do (with merely 3-shot prompt).
>
> We think it is important to know that when it comes to IR there is nothing special in GPT-3 and that BLOOM (or GPT-J) can easily replace it.
>
> Newer models like LLAMA-2 can probably work better, but their training data is typically not document/released/disclosed. You can possibly take MS MARCO and instruction-finetune LLAMA on it. Will the model learn IR? Likely, it will, but it we will learn little about effectiveness of next-token-prediction training.
>
> Finding better OSS models or prompts combination is an interesting problem, but this seems to be a topic of a separate resource-intensive study.
>
> 3) With respect to "It is just a massive empirical exercise showing several things that "worked."". First, we confirmed that prompt-distillation of LLM-s to small models can easily fail to outperform BM25, which is an important piece of knowledge. Yet, we also showed which additional steps can be taken to fix this (without using very large privately trained LLMs and many more prompts). We think that showing which combination of things works and which does not is an insight on its own.
>
> We also find that **only** a small ranking model benefits from in-domain pre-training, which is, in our opinion, is an additional insight. It is not quite clear why this happens and there could be opportunities to improve training of larger rankers as well.
>
> 4) On page 7, we did mention that: "First we carried out an all-domain pre-training without any filtering (i.e., using all queries from all
> collections)." We will add additional clarifications to make sure readers do not miss what all-domain training is.
>
> 5) With respect to consistency checking thresholds: **which values would you recommend to test**? Would it be sufficient to run such as an experiment using, e.g., only MS MARCO?

---

> ### Author Response · Authors · 2023-11-02
> **please, respond to our previous comment**
>
> Dear reviewer,
>
> Please, note that the discussion period has begun. Please, respond to our previous comment.
>
> To reiterate: Our major point is that we did test more than one OSS model. We did test GPT-J (see Table 4) in addition to BLOOM. Because experimentation is rather massive (multiple collection generation and training rankers using three seeds), it is not feasible to do an extensive testing of OSS models, especially, because it is **NOT** required to answer our key research questions. Of course, it is nice to know which open LLM work better in this context, but it seems to be a topic for a separate study. For us it was sufficient to know that we can outperform GPT. Frankly, we thought this would not be possible, but hoped to obtain only "competitive" (but worse) results.

---

> > ### Comment · Reviewer_KGAf · 2023-11-03
> > **Thank You**
> >
> > Thank you for your detailed response and clarifications on the concerns raised in my review.
> >
> > I appreciate your emphasis on the distinction between a replication and reproducibility study. The clarification helps situate your paper in the right context. I also understand TMLR's stance on welcoming both reproducibility and replicability studies.
> >
> > Regarding the use of other OSS models, I acknowledge your reasoning and the fact that you did test with GPT-J, which I might have overlooked in my initial review. The challenge of extensive experimentation, especially considering the volume of data and models, is a valid one. However, I believe highlighting this challenge and your rationale for selecting BLOOM (and GPT-J) more prominently in the paper would be beneficial for the readers.
> >
> > Your point on the effectiveness of next-token-prediction training and the exploration of other models or prompts as a separate, resource-intensive study is well-taken. The insights you provided on which combinations work and which don't, and the nuances on small ranking model benefits from in-domain pre-training, are valuable and indeed contribute to the field.
> >
> > Regarding the consistency checking thresholds, my suggestion would be to experiment with a range of values that can capture both lenient and strict thresholds. An experiment using only MS MARCO or a representative dataset would be a reasonable approach, considering the intensive nature of such evaluations.
> >
> > Lastly, I appreciate your willingness to provide additional clarifications on the all-domain pre-training procedure. Such details would definitely enhance the clarity of the paper for the readers.
> >
> > I look forward to seeing the revised version of your manuscript.

---

### Review · Reviewer_ECET · 2023-09-16

**Summary Of Contributions:**

Previously, the InPars paper showed solid improvement in using LLM to help train ranking models for document retrieval. This paper reproduced and validated the InPars work independently.

More importantly, the paper noted that the original InPars method used larger proprietary models GPT-3, which is hard to verify that the training is not contaminated with training/test data. Larger models also means higher cost of use.

The paper then proposed an improved version of InPars, called InPars-light. The new approach uses open source models instead of GPT-3, the training process of which is well understood by the research community. It uses smaller BERT models than the MonoT5 used by the original InPars paper, and leverage consistency checking to help filter the generated data. Results showed that the open-source models and smaller rankers can produce comparable results than the proprietary model and larger ranker. The paper summarized it into a recipe and performed extensive experiments on MS MARCO dataset to answer the questions regarding model size reduction, effectiveness of the consistency checking, and transfer learning abilities.

**Audience:**

Yes

**Broader Impact Concerns:**

There's no broader impact concern.

**Claims And Evidence:**

Yes

**Requested Changes:**

Section 3.4 Paragraph 3. The "average" run setup is still not clear to the reviewer. The authors said "we first obtained a ...", implying there is a "then..." but isn't.

**Strengths And Weaknesses:**

Strengths.

1) Well written. It's easy to follow, with good examples and well formulated research questions.
2) All research questions are clearly answered with extensive experiments.
3) Cost reduction. Experiments showed that smaller models can achieve the same level of result.
4) Open source. Showed that non-proprietary models can repeat the result, which reduces the cost and risk of data contamination.

Weakness.

The reviewer didn't find any significant weakness in the paper. Please see the requested changes for smaller questions.

---

### Review · Reviewer_TFJB · 2023-10-30

**Summary Of Contributions:**

The main contribution of the paper is a reproducibility study of InPars, a method for training a ranking model, using open-source models (specifically BLOOM & GPT-J). They improve the method in several ways, getting strong results with smaller models than the original paper.

The main idea behind InPars and (the new proposed InPars-Light) is using an LLM with a few-shot prompt to produce synthetic training examples.

They successfully reproduce InPars and show improved and comparable results can be achieved with a better LLM, improved techniques and smaller models.

**Audience:**

Yes

**Broader Impact Concerns:**

None.

**Claims And Evidence:**

Yes

**Requested Changes:**

I don’t have any major requests, just a suggestion based on personal preference: Posing the research questions in the introduction, and leaving them unanswered until later, may leave some readers impatient.

**Strengths And Weaknesses:**

Strengths:
*  It’s a very well-written, clear and well-structured paper.
* They train rankers using multiple seeds and use statistical tests when measuring improvements.
* They perform ablation studies of important parts, e.g. the consistency checking.

Weaknesses:

* Given the strong results with the small models, it would be interesting if better results would be possible with larger models. The paper leaves this open.
* The authors use an anonymous retrieval toolkit. They do however describe experiments that compare with Pyserini and show that those closely match.

---

> ### Author Response · Authors · 2023-11-02
> **we do answer RQs on the second page!**
>
> Thank you very much for the toughtful review. We do answer research questions in the beginning of the paper, but without explicitly mentioning them. We will connect our summary of findings with respective RQs for more readability in the final version.

---

> > ### Comment · Reviewer_TFJB · 2023-11-09
> >
> > Yes, you're right, but the matching wasn't that obvious, at least to me, (and the details are deeper in the paper). But as I said, this is more of a suggestion and as reviewer I'm OK if you're not making any changes based on it.

---

### Decision · Action_Editor_6ogb · 2023-12-14

**Recommendation:** Accept with minor revision

**Comment:**

The reviewers generally commend the paper for its well-written and clear presentation, thorough experimental methodology, and valuable insights into reproducibility and cost-effectiveness of neural rankers. The main contributions involve a reproducibility study of InPars and the introduction of InPars-Light, a more cost-effective modification. While reviewers lean towards acceptance, they suggest a weak opposition due to the perceived lack of novelty. There are recommendations to explore other open-source language models for data generation, delve into consistency checking thresholds, and provide more details on certain procedures. Despite these suggestions, the paper is acknowledged for its rigorous empirical exercise and insights into what works in the context of prompt-based training of neural rankers. Therefore, I recommend accepting the paper with minor revisions to address the concerns raised by reviewers and enhance the clarity of certain aspects.

**Audience:**

At least some individuals in TMLR's audience would likely be interested in knowing the findings of this paper. The paper addresses important questions related to the reproducibility of the InPars method and introduces an improved version, InPars-Light, which uses smaller models and open-source language models. The relevance of the paper to the audience is evident in the positive comments and the recommendation for acceptance, even though there are some suggestions for minor improvements. Overall, the paper's contributions to the understanding of reproducibility in neural rankers and the cost-effectiveness of different approaches make it potentially interesting and relevant to TMLR's audience.

**Claims And Evidence:**

The claims made in the submission are generally supported by accurate, convincing, and clear evidence. The reviewers acknowledge the paper for the thorough evaluation on multiple standard information retrieval datasets. The evidence provided includes statistical testing to validate improvements over baselines, comprehensive evaluation on various datasets, and a detailed breakdown of computational costs and latency measurements. However, there are suggestions from one reviewer to explore certain aspects further, such as experimenting with different consistency checking thresholds and considering other open-source language models for data generation. These recommendations are made to enhance the completeness and robustness of the evidence presented.

---

> ### Author Response · Authors · 2023-12-27
> **re:decision**
>
> Thank you very much for a thoughtful decision: We will upload an updated version soon.